# Lipid-Based Nanotechnology: Liposome

**DOI:** 10.3390/pharmaceutics16010034

**Published:** 2023-12-26

**Authors:** Yanhao Jiang, Wenpan Li, Zhiren Wang, Jianqin Lu

**Affiliations:** 1Pharmaceutics and Pharmacokinetics Track, Skaggs Pharmaceutical Sciences Center, Department of Pharmacology & Toxicology, R. Ken Coit College of Pharmacy, The University of Arizona, Tucson, AZ 85721, USA; jianyanh@arizona.edu (Y.J.); wli@pharmacy.arizona.edu (W.L.); zwang@pharmacy.arizona.edu (Z.W.); 2Clinical and Translational Oncology Program, NCI-Designated University of Arizona Comprehensive Cancer Center, Tucson, AZ 85721, USA; 3BIO5 Institute, The University of Arizona, Tucson, AZ 85721, USA; 4Southwest Environmental Health Sciences Center, The University of Arizona, Tucson, AZ 85721, USA

**Keywords:** liposome, drug delivery, disease treatments

## Abstract

Over the past several decades, liposomes have been extensively developed and used for various clinical applications such as in pharmaceutical, cosmetic, and dietetic fields, due to its versatility, biocompatibility, and biodegradability, as well as the ability to enhance the therapeutic index of free drugs. However, some challenges remain unsolved, including liposome premature leakage, manufacturing irreproducibility, and limited translation success. This article reviews various aspects of liposomes, including its advantages, major compositions, and common preparation techniques, and discusses present U.S. FDA-approved, clinical, and preclinical liposomal nanotherapeutics for treating and preventing a variety of human diseases. In addition, we summarize the significance of and challenges in liposome-enabled nanotherapeutic development and hope it provides the fundamental knowledge and concepts about liposomes and their applications and contributions in contemporary pharmaceutical advancement.

## 1. Introduction

Cancer has brought a critical burden to the economy and society. GLOBOCAN (the World Health Organization’s International Agency for Research on Cancer Global Cancer Observatory) 2020 reported an estimation of 19 million new cancer cases and 10 million cancer deaths occurred worldwide [1]. Currently, cancer treatments are still mainly proceeded by surgery, radiotherapy, and chemotherapy, although gene therapy and immunotherapy have been brought up as novel methods with a higher therapeutic index. However, some challenges remain unsolved even with the advanced therapies, such as low solubility, poor pharmacokinetics, non-specific biodistribution, and systemic toxicities [2,3]. Therefore, targeted delivery of therapeutics to specific sites has been an active area of research in the last couple of decades. Of note, several drug delivery platforms have been reported, and some are being used in clinical settings, including antibody-drug conjugates, polymers, as well as liposomes [4,5,6]. Of those, liposomes are a promising drug delivery vehicle due to their biocompatibility and biodegradability, good stability, as well as the ability to encapsulate both hydrophobic and hydrophilic contents [7]. When the first liposome was described by Bangham et al. in 1964 [8], it had grown to be a great interest in cosmetic, dietetic, and pharmaceutical areas [9,10,11].

Due to the natural properties of liposome, the major components are lipids and fatty acids comprising phospholipids, which can spontaneously self-assemble into a lipid bilayer with an aqueous core. The phospholipid bilayer is similar to the construction of the cell membrane. Therefore, liposomes are considered to be biocompatible and biodegradable [7]. Because of the presence of a lipid membrane and a hydrophilic interior, liposomes can be used to deliver both hydrophilic and hydrophobic molecules. With that, liposomes have been further researched of their benefits as a drug delivery platform (Table 1).

This review will summarize the characterizations of liposomes and elaborate some major liposome preparation methodologies as well as explore the applications of liposomes in different diseases and introduce some of the U.S. FDA (Food and Drug Administration) approved and preclinical- and clinical-testing liposomal nanodrugs. We also discuss recent novel applications and comment on the disadvantages of liposomes. We believe that this review is an attempt to provide a detailed insight in understanding how to prepare liposomes and why liposomes are used as nanocarriers for therapeutic delivery.

## 2. Characterization and Major Components of Liposomes

Several ways can be used to classify liposomes, including size, lamellarity, and method of preparation [12,13]. We define liposomes by their size and lamellarity. These two factors also dominate the drug encapsulation efficiency and ADME (absorption, distribution, metabolism, and elimination) of the drug [7,14,15]. By lamellarity, liposomes can be defined as: a unilamellar vesicle (ULV), with one bilayer membrane; an oligolamellar vesicle (OLV), with 2–5 bilayer membranes; or a multilamellar vesicle (MLV), with five or more bilayer membranes. Furthermore, ULV can be classified by its size, including small unilamellar vesicle (SUV) ranging from 20 to 100 nm; large unilamellar vesicle (LUV) with a size larger than 100 nm; and giant unilamellar vesicle (GUV) with a size bigger than 1000 nm [16]. Generally, ULV is formed by a phospholipid bilayer and an aqueous core. More uniquely, several ULVs with gradually smaller sizes caging inside each other compose the MLV, which resembles an onion, and each lipid bilayer is separated by an aqueous layer [17].

Three dominant components that contribute to the formation, stability, and functionality of liposomes include phospholipids, cholesterol, and polyethylene glycol (PEG).

### 2.1. Phospholipids

Phospholipids, amphiphilic in nature, contain both polar and non-polar groups. In the aqueous solution, phospholipids self-assemble into a bilayer sheet where the headgroups align, forming two lipid layers with their tail groups facing each other [18].

Generally, a phospholipid is composed of hydrophobic fatty acyl chains, a glycerol or sphingosine backbone, and a hydrophilic headgroup. The fatty acid chain takes part in structure functions of phospholipids, also called “building blocks” of the cell membrane [19]. It can differ by length, branch, and degree of saturation of the carbon chain [20]. Phosphatidic acid (PA) is a phospholipid found in cell membranes and acts as a building block for the biosynthesis of other phospholipids (Figure 1). Common phospholipids, including phosphatidylethanolamine (PE), phosphatidylglycerol (PG), phosphatidylcholine (PC), phosphatidylserine (PS), and phosphatidylinositol (PI), are all derivatives of PA [21], and the main differences among these phospholipids are the functional head groups (Table 2). Cationic phospholipids, such as 1,2-di-O-octadecenyl-3-trimethylammonium propane (DOTMA), 1,2-Dioleoyl-3-trimethylammonium propane (DOTAP), and Dimethyldioctadecylammonium (DDAB), are well known for their ability for gene delivery (Figure 2) [22]. Moreover, the most representative of sphingosine-consisting phospholipid is sphingomyelin, which contains an amino alcohol on the backbone [23]. As the main structural and functional component in liposomes, phospholipids play a vital role in the constitution of liposomes, and they also act as a surfactant and help in solubilization and metabolism of the vesicles, as phospholipids can not only maintain the integrity of cellular structure and functions to regulate the passage of molecules, ions, and nutrients in/out of the cells, but also contribute to forming lipid droplets as triglycerides to store energy [24,25,26].

### 2.2. Cholesterol

Cholesterol plays an important role in cell membranes and acts as the precursor to the biosynthesis of several compounds, including bile acid, Vitamin D, and steroid hormones [27,28,29]. Structurally, cholesterol is a rigid molecule with a steroid skeleton of four fused rings, three six-member rings, and one five-member ring (Figure 3). Stereochemistry and oxidation states changes on the fused rings, hydrocarbon side chains, and the functional groups will lead to the biosynthesis of the compounds as mentioned above. Additionally, within the phospholipid bilayer, cholesterol significantly influences membrane fluidity, permeability, and stability by interacting with the nonpolar tails of phospholipids and their polar headgroups. The interaction of cholesterol involves binding its nonpolar portion to the hydrophobic tails of phospholipids while its polar head group associates with the hydrophilic headgroups of phospholipids [30,31]. Consequently, cholesterol plays a dual role in contributing to the formation of phospholipid bilayers and impacting the properties of liposomes. Moreover, it impacts the rigidity, strength, and permeability of the lipid membrane by modulating the ratio of phospholipids to cholesterol within the bilayers.

### 2.3. Polyethylene Glyco

Polyethylene glycol (PEG) forms through the polymerization of ethylene glycol, ethylene oxide, or diethylene glycol in the presence of water and a catalyst under pressure [32,33]. The chemical structure of PEG generally comprises OH-[CH_2_-CH_2_-O]n-H. Being hydrophilic in nature, PEG can be covalently attached to the surface of liposome to impart the “stealth” effect [34,35] via educing protein binding and opsonization as well as diminishing degradation by metabolic enzymes. This occurs because PEG prevents the approach or recognition by blood proteins, proteolytic enzymes, and antibodies during circulation, thereby enhancing the therapeutic index in comparison to the raw drugs [36,37]. Notably, PEGylated liposomes have been shown to increase in vivo circulation time and improved formulation stability [38]. This extension of circulation time and enhancement of stability are crucial factors in improving the performance of liposomal drug delivery systems.

### 2.4. Major Methods of Liposome Preparation

Several ways are developed for liposome preparation, both conventional and novel techniques. Different methods can affect the properties of liposomes, including size, lamellarity, and encapsulation efficiency [12,13,14].

#### 2.4.1. Thin Film Hydration

When the thin film hydration method (Figure 4), also called the Bangham method, was first described in 1967, it was used as the most common and simplest technique for liposome synthesis in laboratory scale [39]. In this method, the lipids and amphiphilic molecules are mixed in the organic solvent, as the liposome formation precursor, which is then transferred to the round-bottom flask. Following the solvent evaporation by rotary evaporator, a thin lipid film on the bottom of the flask is formed. A freezing-drying procedure, typically conducted on hydrated lipids, can also be utilized to ensure the complete the removal of solvents. The film is then hydrated with an aqueous solution that contains drugs. This step facilitates the encapsulation of drugs into the lipid thin film, and subsequent hydration triggers the formation of liposome. It is also crucial that the temperature of the hydration buffer remains above the gel–lipid phase transition temperature ™ [13]. Furthermore, several other critical factors affecting liposomal characteristics include the volume of the hydration solution and the rate of hydration [12,40]. A large volume of the hydration solution tends to generate a great number of multilamellar vesicles (MLVs) with a more heterogeneous size distribution. Additionally, MLVs will be formed if rehydration after freeze-dry is too energetic, in another word, vigorous swirling [41], while a gentle rehydration will form GUVs [40].

Therefore, size reduction methods are needed to generate homogenous SUVs. Two size reduction methods are widely used, including sonication and multiple extrusions. Sonication is mainly applied by probe sonication or water bath sonication. Size distribution of the liposomes can be determined by the frequency ultrasonic waves and the duration of the sonication [42]. A jacked extruder can also be used to reduce the size and generate homogenous size liposomes by multiple extrusions through a polycarbonate membrane. Size distribution of the liposomes is dependent on the number of extrusions and the size of the polycarbonate membrane pores [43,44]. Despite the advantages, the thin film hydration method also has its drawbacks, such as low aqueous core entrapment and low drug encapsulation efficiency [13,16].

#### 2.4.2. Reverse Phase Evaporation

The reverse phase evaporation technique tends to form intermediates of invert micelles or a water-in-oil emulsion, where the encapsulated drugs are stocked in the water phase and lipids comprise the organic phase, which contribute to the liposome formation (Figure 5) [16]. Reverse phase evaporation can generate a mixture of LUVs and MLVs, resulting in the entrapped aqueous phase. This outcome significantly enhances drug loading capacity and allows for the entrapment of larger molecules, such as proteins or nucleic acids [7]. This method is similar to the thin film hydration technique. Starting with mixing the amphiphilic phospholipids and other lipids (e.g., cholesterol) together in the organic solvent and following with evaporation by rotatory evaporator, the thin film is formed in the round-bottom flask. Different from thin film hydration, here, the thin film is redissolved in the organic phase, usually by an organic solvent consisting of diethyl ether and/or isopropyl ether. An aqueous solution containing the drug of interest is then added into the redissolved lipid phase. Following the sonication, it yields a two-phase system, where the liposome is obtained. The organic solvent is then removed under low pressure evaporation, and the conversion system leads to form a viscous gel. The remaining solvent can be removed by dialysis, centrifugation, or passing through a Sepharose gel column [16,45]. For size reduction purposes, multiple extrusions can then be made on the final product. With respect to the desired size distribution of the liposome, it can be determined by the pore size of the polycarbonate membrane and numbers of extrusions [44,45]. The potential drawbacks of the reverse phase evaporation technique lie in the remaining trace elements from the incomplete removal of organic solvent in redissolved liposome, which can interrupt the chemical or biological stability of the drug-laden liposomes [45].

#### 2.4.3. Solvent Injection

From its name, solvent injection is conducted by injecting the phospholipids dissolved in the organic phase into a drug-containing aqueous solution, leading to the formation of liposomes (Figure 6). In this technique [47], lipids and amphiphilic molecules are dissolved in an organic solvent with low boiling temperature. The mixture is then injected into a drug-containing aqueous solution at a constant and higher temperature than the boiling point of the mixture. By this way, the organic solvent can be evaporated, and the liposome vesicle will be formed. Typically, solvent injection will produce LUVs. Alternatively, if the organic solvent has a higher boiling point than the aqueous phase, then after injection, the organic solvent can be removed by dialysis, centrifugation, or passing through a Sepharose gel column [48,49].

The limitations of the solvent injection technique include: high poly-dispersity indexes (PDI), which indicates a heterogeneous size distributions [47], and incomplete removal of the organic phase, which is harmful for liposome construction after formation.

#### 2.4.4. Detergent Removal

The detergent removal method, also called detergent dialysis, uses detergent micelles to produce liposomes. Some detergents can solubilize lipids at their critical micelle concentrations, including cholate, alkyl glycoside, and Triton X-100 [7]. In this method, detergent micelles are added into the drug-containing aqueous solution. The detergent is then removed by dialysis or size exclusion gel chromatography [13,50]. When the detergent is removed, the micelles tend to favor forming liposomes and eventually generate LUVs. The disadvantage of this technique can be emphasized by high PDI.

#### 2.4.5. Micro Hydrodynamic Focusing

The micro hydrodynamic focusing method was first described by Jahn et al., and it was used to form the monodisperse liposome [51]. In this method, two kinds of small microfluidic channels have been used, and both channels have varying diameters up to 500 μm (Figure 7). A flow of phospholipids in an organic phase occur in the center and cross with the perpendicular streams of aqueous buffer containing the drugs. The crossflow results in diffusive mixing and the formation of lipid vesicles. By using this technique, the liposome size distribution and encapsulation efficiencies can be controlled by the aqueous buffer-to-organic phase flow rate ratio, diameter of the microchannels, and concentration of the lipids in the organic solvent [52]. The micro hydrodynamic focusing method has most of the benefits for liposome preparation. Compared with other techniques, this microfluidic injection method is convenient, quick, and affordable, and it can be easily adopted in any biomedical or pharmaceutical research laboratory [53].

### 2.5. Major Methods of Liposome Characterization

The efficacy of a liposome formulation is defined by several key aspects including size distribution, zeta-potential, stability and drug leakage, and phase transition temperature.

#### 2.5.1. Size Distribution and Zeta-Potential

Size distribution and zeta-potential are properties affected by the liposome preparation techniques and compositions, playing critical roles in the liposomal therapeutic delivery [55,56]. Both size distribution and zeta-potential can be measured by dynamic light scattering (DLS), in which the size distribution of liposomes indicates the diameter and polydispersity index (PDI), and zeta-potential emphasizes the overall charge that the liposomes in a particular medium. [57,58,59]. Liposome sizes range from very small (0.025 µm) to large (2.5 µm), typically designed to be <400 nm to enable passive accumulation of liposomes within the tumor microenvironment through the enhanced permeation and retention (EPR) effect [7,60]. The measurement of PDI determines the uniformity of the liposome size, where the smaller the PDI, the more uniform the liposome will be [61]. Liposome surface charge can decide its targeting. Liposomes with positive surface charge facilitate the absorption into tissues, whereas those with a slightly negative charge appear to prolong the circulation time by efficiently minimizing the protein binding and opsonization. Thus, we can adjust the surface charges on liposomes to potentially improve in vivo targeting efficacy [62,63]. Additionally, liposomes with neutral surface charge tend to aggregate [13].

#### 2.5.2. Stability and Drug Leakage

Liposome stability is a critical index that impacts its potential efficacy and utility in clinic. The physical and chemical instability of liposomes can lead to side effects and efficacy reduction [64,65]. Physical stability refers to the ability of liposomes to maintain their structural integrity over time. It is crucial for their storage, handling, and successful delivery to the target site [66]. The stability of a liposome formulation is typically analyzed by measuring size distribution and zeta-potential at multiple timepoints, such as days, weeks, or months, and assessing drug leakage [13]. If liposomes experience physical instability, they may undergo aggregation, fusion, or leakage of their contents, which can result in a loss of drug efficacy or altered pharmacokinetics.

Chemical stability refers to the ability of liposomes to resist degradation and maintain the integrity of the encapsulated drugs or therapeutic agents. Chemical instability in liposomes could arise from the degradation mechanisms of oxidation and hydrolysis, etc. Oxidation is highly likely to occur as a result of the presence of free radicals in fatty acids, which serve as intrinsic components. Within this mechanism, unsaturated fatty acids exhibit greater susceptibility compared to their saturated counterparts [67]. If liposomes undergo chemical instability, the active compounds may degrade, lose their potency, or undergo undesirable chemical reactions, leading to reduced therapeutic effectiveness [68]. Additionally, instability of liposomes induced by the chemical nature of the encapsulated drugs also remain a concern in drug delivery. The interactions between drugs and phospholipids can disrupt the chemical stability of liposomes due to various drug-related factors, including hydrophobicity/hydrophilicity, pH sensitivity, and chemical reactivity [64,69,70]. Hence, understanding the inherent chemical properties of the delivered drug is crucial in designing liposomal formulations that minimize leakage and improve drug retention within the vehicles.

The propensity of hydrogen bond formation in liposomes is crucial for structural stability. Hydrogen bonds serve as a pivotal factor in maintaining the structural integrity of liposomes [71]. These bonds are conducive to stabilizing the phospholipid bilayer, which constitutes the basic structure of liposomes, by enhancing the interaction between lipid molecules. This could prevent a breakdown and sustain the integrity of the vesicle structure. Moreover, it also plays a vital role in membrane permeability [72]. Depending on the types or strength of hydrogen bonds present, the liposome membrane displays varied degrees of permeability. Consequently, it is possible to regulate the encapsulation and subsequent drug release within the liposomes via modulating the hydrogen bonding.

#### 2.5.3. Phase Transition

Phase transition temperature (Tc) of phospholipids also stands as a critical parameter influencing the fluidity of liposomes within the lipid bilayer. This transition, from a rigid gel phase to a more fluidic liquid crystalline phase, significantly impacts the permeability and aggregation tendencies of the liposome [69,73]. Moreover, the Tc of phospholipids is contingent upon on the lengths and types (saturated and unsaturated) of their fatty acid chains [74]. Consequently, monitoring the fluidity of bilayers or liposomes can be achieved by selecting different phospholipids.

Additionally, lipid aggregates undergo two distinct types of phase transition: thermotropic and lyotropic [75].

Thermotropic phase transitions occur when the temperature changes, leading to alterations in the organization and composition of lipid structures. Lipid molecules can transition between various states, such as from a structured and rigid gel phase to a more fluid and less ordered liquid crystalline phase as temperature increases. These changes relate to the packing and mobility of lipid molecules within the assemblies as temperature fluctuates [75,76].

Changes in solvent concentration, such as water, within a system containing lipids or amphiphilic compounds lead to lyotropic phase transitions. These transitions involve restructuring and altering the structure of lipid aggregates, such as micelles or bilayers, in response to changes in solvent concentration. For example, fluctuations in water concentration prompt lipid molecules to adopt various organized arrangements like lamellar structures (sheets), hexagonal patterns, or micelles. These modifications occur due to interactions between the lipids and the solvent molecules [77].

#### 2.5.4. Fluorescence Microscopy

Fluorescence microscopy techniques enable multi-color imaging of individual liposome, providing a more quantitative and comprehensive understanding of their biochemical properties [78]. Among these techniques, two commonly utilized methods include laser scanning confocal fluorescence microscopy (LSCFM) and total internal reflection fluorescence microscopy (TIRFM). LSCFM, known for its high resolution, optical sectioning, and 3D imaging capabilities [79,80], serves to characterize liposomes and observe their structure, behavior, and interactions with high precision [81,82]. In contrast, TIRFM is well-known for its accessibility and high sensitivity in probing biomolecules properties [83]. When characterizing liposomes, TIRFM presents unique advantages by allowing observations of membrane-related phenomena and interactions at the liposome interface [84,85]. Both LSCFM and TIRFM play pivotal roles in unraveling the liposomal structure and their interactions, contributing significantly to our understanding of the liposomal delivery platforms.

#### 2.5.5. Fourier Transform Infrared Spectroscopy

Fourier transform infrared spectroscopy (FTIR) is a potent analytical technique extensively utilized in liposomes research, including composition assessment, structural analysis, interaction studies, and stability evaluation [86,87,88]. Different functional groups in lipids have unique peaks in the FTIR spectrum, enabling the straightforward determination of lipid composition and ratios within liposomes. Changes in specific absorption bonds will be indicated as interactions between lipid and encapsulated molecules, such as hydrogen bonding, while also serving as a mean to access liposome stability by detecting alteration because of oxidation or degradation. Thus, FTIR provides invaluable insight into the liposome characterization, contributing significantly to the optimization of liposome-based drug delivery platforms.

### 2.6. Stimuli-Responsive Liposomes

Stimuli-responsive liposomes are specifically designed to release their payloads in response to specific external stimuli. Many environmental stimuli have been explored including pH, temperature, and enzymes.

#### 2.6.1. pH-Responsive Liposomes

pH-responsive liposomes are specialized nanocarriers that are responsive to alternations in the acidity or alkalinity of the surroundings. These liposomes are engineered to maintain their stability at certain pH and then can rapidly dissociate once exposed to a different pH environment [89]. Normally, most of the components in pH-responsive liposomes are similar to other liposomal drugs, except for the pH-sensitive component. Zhao et al. developed a pH-responsive liposome platform for co-delivery of PLK-1 specific siRNA and docetaxel using a dual pH-sensitive peptide sequence, composed of three units: a cell-penetrating domain (polyarginine), a polyanionic shielding domain (ehG)n, and an imine linkage [90]. During the blood circulation, the liposomal formulation remains inert until reaching the acidic tumor microenvironment, and then its contents release is initiated by the breakage of the imine bonds within the pH-responsive peptide under acid-catalyzed hydrolysis.

Additionally, pH-responsive liposomes have been investigated for the delivery of various therapeutic agents, such as proteins, nucleic acids, and chemotherapeutics drugs [91,92].

#### 2.6.2. Temperature-Responsive Liposomes

Temperature-responsive liposomes can alter their original structures or behavior in response to varied temperature. These liposomes are designed to maintain their stability at one temperature and then undergo conformational changes at a different temperature threshold.

Zhao et al. investigated a temperature-responsive liposome based on the insertion of the ion pair of polyethyleneimine (PEI) and (phenythio)acetic acid (PTA) into the liposomal bilayer [93]. This ion pair, formed by electrostatic interaction between PEIs and PTAs, possesses amphiphilic properties, facilitating self-assembling in the aqueous solution [94]. Being a compatible molecule, the amphiphilic ion pair contributes to the creation of DOPE liposomes, with the PTAs’ phenyl head group inserting into the bilayer while PEI’s chain fills the space among the phospholipid headgroups. As the temperature rises, the ion pair loses its amphiphilicity due to the hydration of the phenyl group on PTAs [95]. This enables the detachment of the ion pair from DOPE liposomes, destabilizing the liposomes and consequently releasing the payload.

Temperature-sensitive liposomes provide the advantage of precise control over drug release triggered by temperature change, which has been widely applied in various cancer treatments [96,97].

#### 2.6.3. Enzyme-Responsive Liposomes

Enzyme-sensitive liposomes are designed to release their encapsulated payloads in the presence of specific enzymes. The controlled release of their contents is triggered by the exposure or recognition of particular enzymes.

Chasteen et al. explored enzyme-responsive liposomes by modifying the N-acylated DOPE into the liposomes [98]. Exposure to specific temperature (55 °C to 65 °C) in the presence of palladium, as a catalyst for biorthogonal chemistry [99], will generate triggered-release liposomes. This involves crafting chemically caged polar lipids capable of transitioning from a structure that maintains a liposomal membrane to one that disrupts the bilayer.

Enzyme-responsive liposomes show potential for targeted drug delivery in various medical fields, as they could be easily tailored to respond to different enzymes associated with specific diseases for precise therapeutic delivery [96,100].

## 3. Pharmaceutical Applications of Liposomes

Owing to its biocompatibility, biodegradability, nontoxicity, and favorable physical properties for convenient modifications of surface charge and its size, since the 1990s, there have been more than a dozen U.S. FDA-approved liposomal or lipid-based nanodrugs (Table 3) with numerous more under preclinical and clinical development.

### 3.1. Anti-Cancer

Cancer is a disease which is flourishing rapidly throughout the world. The ultimate goal of cancer therapy is to destroy all the malignant cells. Conventional chemotherapy, as one of the most common cancer treatments, employs cytotoxic agents that target rapidly proliferating cells, especially like cancer cells [114]. For instance, anthracyclines, including Daunorubicin, Doxorubicin, Epirubicin, Idarubicin, Mitoxantrone, and Valrubicin, are used as chemotherapeutic agents for treatment of numerous types of cancers [115]. Yet, chemotherapies have been associated with severe unwanted systemic toxicities, off-target effect, and rapidly emerging drug resistance [116]. Most of the chemotherapeutic drugs are not selective to cancer cells, which indicate that they not only target cancer cells, but also can be randomly distributed to healthy organs. As reported, detrimental effects to the central nervous system (CNS) are recognized as cognition dysfunction during chemotherapy for a non-CNS cancer [117]. Furthermore, chemotherapeutic drug resistance of malignant cells builds another bottleneck for cancer chemotherapy. Several factors are attributed to the drug resistance, including the heterogeneity of the tumor cell population [118], the tumor microenvironment, and the limited ability of the drug to penetrate tumor tissue to reach the potential lethal concentration for all tumor cells [119]. To overcome these challenges, various drug delivery systems have been developed including viral [120,121] and non-viral vectors such as liposomes [122].

#### 3.1.1. Doxil

The liposome has been the most successful in therapeutic delivery as evidenced by numerous FDA-approved liposomal nanodrugs (e.g., Doxil, DaunoXome, Depocyt, Myocet, Mepact, and Onivyde, etc.) for diverse diseases management (e.g., cancers). Doxil, the first FDA-approved nanodrug delivery system using pegylated liposomes to encapsulate doxorubicin, consists of three major components: the high-transition-temperature (T_m_) phospholipid hydrogenated soy phosphatidylcholine (HSPC; T_m_ 52.5 °C); cholesterol; and *N*-(carbonyl-methoxypolyethylene glycol 2000)-1,2-distearoyl-*sn*-glycero-3-phosphoethanolamine sodium salt (MPEG-DSPE) [101]. The pharmacokinetics of Doxil were analyzed by Gabizon et al. [123]. The clinical result substantiates that the longevity of liposomes in circulation prolongs the drug’s circulation time compared to free drugs. Consequently, this enhances drug accumulation in malignant tissues, resulting in improved anticancer efficacy [124].

#### 3.1.2. Onivyde

Onivyde, also known as an irinotecan liposome injection, is used for patients with metastatic adenocarcinoma of the pancreas with cancer progression after the gemcitabine-based therapy, usually in combination with fluorouracil and leucovorin [125]. The Onivyde liposomal vesicles comprise three key components: distearoylphosphatidylcholine (DSPC), cholesterol, and methoxy-terminated polyethylene glycol (MW2000)-distearoylphosphatidylethanolamine (MPEG-2000-DSPE) [126]. The efficacy and safety of Onivyde were evaluated in a global, randomized, open-label NAPOLI-1 clinical trial involving patients with metastatic pancreatic cancer who experienced disease progression after gemcitabine treatment [127]. The clinical results confirmed that liposomal irinotecan, Onivyde, significantly extends the lifespan of patients compared to free drugs.

#### 3.1.3. Liposome-Peptide Conjugated Drugs

Peptides play a critical role in genes and drugs delivery, classified into two types: cell-penetrating peptides and cell-targeting peptides (Figure 8) [128]. Peptides exhibit advantageous properties, being biocompatible and well-tolerated, with modifiable features such as hydrophobicity, charge, solubility, and stability [129]. While most of the cell-penetrating peptides are cationic peptides and possess the ability for cellular uptake without inducing cytotoxicity, they lack selectivity and receptor-dependence, thereby limiting tissue specificity and tumor targeting [130]. As the need for enhanced peptide targeting and selectivity emerged, liposomes have been introduced as a delivery platform, forming an engineered combination known as liposome–peptide conjugates [129,130]. These conjugates showcase remarkable performance improvements in cellular uptake, tumor penetration, extended circulation time, and enhanced site-specific targeting, surpassing both liposomal drugs and free drugs [131]. Recent studies have highlighted the use of a chimeric peptide, RIPL, formed by combining cell-specific (IPL, IPLVVPLC) and cell-penetrating (R8, RRRRRRRRC) peptides. This RIPL peptide was conjugated with a liposome, creating RIPL peptide-conjugated liposomes [132,133]. The IPL component in this strategy targets hepsin, a protein overexpressed in cancer cells, exhibiting high affinity and selectivity to IPL peptides [134]. Consequently, this approach significantly enhances cellular uptake and strengthens selective binding with the RIPL–liposome conjugates [135].

### 3.2. Anti-Fungal

There are two forms of fungi existing in nature, yeasts and molds [137]. Most fungi do not live dependent on animals or human beings. Yet, some groups are exterior pathogens in humans, such as *Candida* spp., *Aspergillus* spp., *Cryptococcus* spp., *Fusarium* spp., Mucorales, and endemic mycosis [138], and these cause superficial, subcutaneous, or systemic infections. Additionally, a severe, systemic fungal infection with yeasts or molds is clinically described with invasive fungal infection. Although some infections, like superficial infections, are not life-threatening, the consequences could be severe and affect the patient’s quality of life [139]. On the other hand, in immunocompromised patients, for example, bone marrow and organ transplant patients, systemic fungal infections are associated with high mortality rates [140,141].

#### 3.2.1. Amphotericin B and Ambisome

Amphotericin B is one of the most widespread therapeutic polyene antifungals [142]. According to the Infectious Diseases Society of America (IDSA) [143] and the European Confederation of Medical Mycology (ECMM) [144], Amphotericin B is still recommended as first line treatment polyene antifungals used for severe cryptococcosis, disseminated histoplasmosis, and mucormycosis. However, a number of studies show that Amphotericin B treatments of systemic mycosis caused by species such as *Aspergillus terreus* [145], *Scedosporium* spp. [146], and *Candida auris* [147] are not always effective, which results from the intrinsic or acquired drug resistance [148]. Moreover, the intrinsic host toxicity of Amphotericin B is another clinical concern [148]. Herein, the maximum tolerance dose of Amphotericin B deoxycholate is assigned within the range of dose-related toxicity, and the affinity of Amphotericin B to fungal ergosterol (K_d_ = 6.9 × 10^5^) is 10-fold higher than its affinity to mammalian cholesterol (K_d_ = 5.2 × 10^4^), yet Amphotericin B can non-selectively target mammalian cell membrane and disrupt its structure [149,150]. Nevertheless, increasing exposure of Amphotericin B to renal cells can cause nephrotoxicity [150]. As a result, the treatment choice of Amphotericin B is often limited by its intrinsic drug resistance, and dose reduction is necessary to avoid nephrotoxicity. Therefore, to overcome these drawbacks, lipid-based formulations have been developed to enhance the therapeutic index and lessen the toxic complications.

To date, several liposomal formulations for anti-fungal infections have been approved by the FDA, including Abelcet, Ambisome, and Amphotec. Ambisome was developed by Astellas Pharma USA for the treatment of serious, life-threatening fungal infections, and also for Amphotericin B intolerance or renal-impaired patients who were infected with invasive systemic infections caused by *Aspergillus*, *Candida*, or *Cryptococcus* [101]. Structurally, the lipid bilayer of Ambisome is composed of hydrogenated soy phosphatidylcholine (HSPC), cholesterol, 1,2-distearoyl-sn-glycero-3-phosphoglycerol (DSPG), and Amphotericin B [151]. Additionally, Ambisome showed its potential in prolonged circulation time and high circulation plasma levels at 37 °C due to the lipids presented therein, including HSPC and DSPC [152].

Preclinical data reported by Adler-Moore et al. illustrated that Ambisome showed significant reduction in toxicity and improved therapeutic index in animal models for the treatment of systemic fungal infections [153]. Walsh et al. [154] established an open-label, multidose pharmacokinetic study with 36 patients who were assigned one of four different dosage cohorts of Ambisome, 1.0 mg/kg (N = 8), 2.5 mg/kg (N = 8), 5.0 mg/kg (N = 12), and 7.5 mg/kg (N = 8), to determine the safety, tolerance, and pharmacokinetics of Ambisome. Walshe et al. observed that continued administration of Ambisome resulted in more sustained plasma levels and decreased total body clearance. In other words, Ambisome increased the circulation time.

#### 3.2.2. Nystatin and Nyotran

Like Amphotericin B, Nystatin is a polyene antibiotic. However, due to its systemic toxicity and low intestinal permeability, the therapeutic application of Nystatin has been limited to topical use in mucocutaneous (oral) and cutaneous (vaginal) forms of candidiasis [155].

To overcome these limitations, lipid-based nanotechnologies have been applied to Nystatin as a multilamellar liposome, known as Nyotran, used in treating systemic fungal infections [156]. With the liposomal formulation, Nyotran shows reduced toxicity, improved pharmacokinetics, and better tolerability [157]. Another clinical report also stated that Nyotran was active in some patients in which Amphotericin B treatment failed [158].

#### 3.2.3. Inhaled Liposomal Antimicrobial Medications

Besides the most common routes of liposomal drug delivery, such as oral, typical, and parenteral, the use of inhaled liposomal dosage form for treating respiratory diseases and/or infections has been used increasingly in clinical practice [159]. Inhaled liposomal dosage forms offer potential advantages for the treatment of respiratory infections including targeted delivery, reduced systemic toxicity, improved efficacy, and minimized side effects [160,161,162].

Amphotericin B, as the common treatment for pulmonary fungal infections, is limited by high mortality in achieving the minimum inhibitory concentration in the lung [163]. To address these challenges, clinical studies have explored the administration of a liposomal amphotericin B parenteral formulation alone or with amphotericin B deoxycholate through nebulization [164]. Nebulization is the method that converts medications into fine mist for inhalation, which has shown promise in enhancing the delivery of amphotericin B and combating pulmonary fungal infections. Apart from the nebulization, pressurized metered-dose inhalers (pMDIs) and dry powder inhalers (DPIs) are also being used for drug delivery, but each device requires different formulations to ensure successful drug delivery into the lung [165,166,167].

### 3.3. Pain Management

Acute pain mostly happens following tissue damage associated with surgery, also known as acute postoperative pain, and chronic pain would persist during the healing process for at least three months after the surgery [168]. Additionally, chronic pain could produce an enormous financial burden for the patients [169]. While chronic pain is not life-threatening, it may have a lasting impact on functioning and influence the quality of life of the patients.

An ideal postoperative pain management should use a multidisciplinary approach to interfere with different pain propagation and perception mechanisms [170]. Moreover, an effective pain management control method is to shorten the inpatient time, avoid opioid dependence or addiction, and reduce the mortality [171]. In addition, regional and local anesthesia play an important role in postoperative pain control, as they block the afferent neural stimuli from the surgical area in order to reach the effective analgesic effect [172]. Local anesthetics, such as bupivacaine, provide more successful pain control than opioids and are widely used for preemptive infiltration during the postoperative period with prolonged duration of action [173]. Moreover, catheter delivery systems are common techniques for continuously administering local anesthetics to prolong the duration of analgesia [174,175]. However, the value of these systems is limited by the cost of the equipment, the difficulty to maintain the correct position of the catheter, and the additional trainings and skills required for clinicians [176]. Thus, the development of novel, long-acting local anesthetics, like liposomal bupivacaine, is potentially important in postoperative pain management.

#### 3.3.1. Exparel

Two FDA-approved liposome formulations (DepoDur and Exparel) have been used for pain management. Exparel is a multivesicular liposomal formulation of bupivacaine being developed for wound infiltration in patients with hemorrhoidectomy and bunionectomy [177]. Exparel is composed of dierucoylphosphatidylcholine (DEPC), which is a novel phospholipid excipient, and other lipid components, including DPPG, cholesterol, and tricaprylin.

The efficacy of liposomal bupivacaine was analyzed by Davidson et al. in eight healthy volunteers [178]. Davidson et al. reported that liposomal bupivacaine showed no reduction in consumption compared to plain bupivacaine with no side effects of local anesthetics. Strikingly, they found that the terminal half-life of liposomal bupivacaine (1294 ± 860 min) after IV administration was around 10-fold longer than that of the plain bupivacaine (131 ± 58 min). Therefore, liposomal formulation imparts significant benefits of prolonging the circulation time of bupivacaine after administration compared with plain bupivacaine.

#### 3.3.2. Liposomal Cannabidiol

Cannabidiol (CBD), a phytocannabinoid discovered in 1940, can be used to treat a number of diseases, such as Alzheimer’s disease, Parkinson’s disease, and chronic pain [179,180,181]. The traditional form of CBD has low oral bioavailability and off-targeting effects, thus impeding its optimal therapeutic index [182]. Nanocarriers have been used for the targeted delivery of various phytocompounds, including CBD, and can improve the stability of phytocompounds, enhance bioavailability, and increase solubility and permeability [183]. A study has shown that liposomal CBD improved bioavailability for pain management in dogs [184]. Another human safety study consolidated the tolerability and safety profile of liposomal CBD to explore its therapeutic delivery further in more clinical studies [185].

### 3.4. Vaccination

Conventional or classical vaccines are based on the use of whole or killed bacteria or viruses to mimic their natural interaction with human immune systems [186,187]. Vaccines remain the most cost-efficient way to defend infectious diseases. Nonetheless, several challenges are yet to be solved, such as the identification of the antigen candidates, ability to induce appropriate immune responses for protection, cross-protection against different strains of pathogens, and route of administration [188]. In vaccine development, the ability of initiating the innate and adaptive immune responses is essential. To elicit a sufficient immune response against the antigens, choosing the appropriate immunostimulatory molecules (e.g., adjuvants) and the efficient delivery platform matters. The adjuvants could not only help prolong the exposure time of the vaccine molecules to the antigen-presenting cells (APCs) but could also interact with APCs and trigger immune responses by themselves [189,190]. Liposomes were first investigated as vaccine adjuvants and a delivery platform in 1974 [191]. Due to their flexibility and versatility, immuno-stimulation induced by liposome-carried vaccines can be modified by various factors including liposome composition, size and homogeneity, charge, and location of antigens and/or adjuvants [192]. It is well noted that the versatility of liposomes in cargo selection plays a pivotal role in vaccine delivery system [193]. Water-soluble antigens such as proteins, peptides, and nucleic acids are encapsulated in the aqueous core of the liposomes, while the lipophilic substrates such as adjuvants, glycolipids, and lipopeptides are entrapped in the lipid bilayers of the liposomes. The antigens could also associate with the surface of the liposome by absorption or covalent binding [194]. Regardless of where the antigens are present (in/on liposomes), the immune responses can be induced by the liposomes, which are phagocytosed by the macrophage and the antigens are processed and presented on the macrophage surface with either the MHCI (major histocompatibility class I) complex if antigens end up in the cytoplasm or the MHCII if antigens end up in the lysosomes. Consequently, the antigen peptides on the MHC complex are recognized by the cytotoxic T lymphocytes (CTLs) and bind to the T cells. Specific cytokines are secreted from the T cells, interacting with B cells and then stimulating B cells to produce antibodies [195].

#### 3.4.1. Liposomal Vaccines

Liposome-based vaccines have been practiced for delivery of several antigens for different disease preventions, including Ag85B-ESAT-6 (H1 antigen) and Ag85B-ESAT-6-Rv2660c (H56 antigen) for tuberculosis vaccines [196,197], Epaxal (hepatitis A antigen) for hepatitis A virus vaccine [198], and Inflexal V (hemagglutinin and neuraminidase from inactivated influenza) for influenza virus vaccine [198].

#### 3.4.2. Lipid-Based mRNA Nanovaccines

Due to the global pandemic of COVID-19, the development of therapeutic and prophylactic options for SARS-CoV-2 (severe acute respiratory syndrome coronavirus 2) have been racing. Two mRNA lipid-based nanovaccines targeting SARS-CoV-2 (Comirnaty and Spikevax, developed by Pfizer/BioNTech and Moderna, respectively) have been approved by FDA to control COVID-19 [112,199,200,201]. While mRNA was found to be safe due to its physiochemical properties such as hydrophilicity and negative charge, it has low passive diffusion efficiency through the plasma membrane [202]. Moreover, free mRNA is degraded quickly in the body, and the short half-life can barely allow it to reach its proposed efficacy [203]. Therefore, mRNA delivery efficiency to the target sites determines its ultimate therapeutic effect. Liposome-associated nanotechnology has shown to protect mRNA from degradation and extend the circulatory half-life, as well as enhance the vaccination effects. Thus, liposome/lipid-based nanoplatforms represent promising strategies for improved mRNA vaccine delivery.

## 4. Discussion and Outlook

The pharmaceutical applications of liposomes are not limited to what have been mentioned above. Liposomal drugs have also been used for photodynamic therapy [204], bacterial infections [205], and cardiovascular diseases [206]. In addition, liposomes have been explored for nanotechnologies as signal enhancers in medical diagnosis [207], solubilizers for various ingredients, and penetration enhancers in cosmetics [55].

Because of the unique characteristics of liposomes, they have also been developed as carriers for brain delivery of bioactive constituents and used for treatments of various central nervous systems disorders such as Alzheimer’s disease [208], ischemic disease [209], and Parkinson’s disease [210,211]. However, to fulfill their clinical translation, the liposomal formulations are required to undergo additional studies to further prove their effectiveness, such as to evaluate the combinations of bioactive molecules, measure the dosage of bioactive molecules administered, and perform assessment in patients with different central nervous system disorders [212].

The liposome exhibits various advantages, such as reducing the side effects, improving the pharmacokinetics, and enhancing the delivery efficiency to target sites as compared to free (unentrapped) drugs. However, liposomes still face some challenges. One critical issue is drug leakage from the liposome during the circulation before it is navigated to the tumor site [213]. Unwanted leakage would not only yield suboptimal circulation times but also release the cytotoxic agents prematurely, damaging the healthy organs/tissues [213]. The primary cause of liposomes’ drug leakage is serum proteins, such as lipoproteins, which can interrupt the integrity of liposome bilayers [213,214]. Moreover, other factors could also impact the stability of liposomes, including types of phospholipids, drug-to-lipid ratio, and liposome compositions [13,61]. One approach to stabilize liposome bilayer is to modify its surface with polymers, typically PEG, which is known as polymer-stabilized liposomes [215]. This strategy is to covalently link the polymers onto the hydrophilic headgroups of the lipids or physically absorb the polymers into the surface of the liposomes. The polymers can either act to repel serum proteins in the plasma [216] or to adjust the drug release rate [217]. Furthermore, off-targeting has generated an unignorable hit to liposomes as they not only accumulate at the target site (e.g., tumors), but also stagger in normal tissues such as in the liver, kidney, and spleen, yielding non-specific side effects [218,219]. Various strategies have been explored to enhance the targeting efficiency of liposomes, such as functionalizing their surface with tumor-specific ligands (small molecules: folate, biotin, vitamin A, etc.; peptides: iRGD, RIPL, NZX, etc.; antibodies; aptamers, etc.) [132,220,221,222,223,224,225,226,227]. However, as of now, there is no FDA-approved ligand-decorated liposomal nanomedicine. Hence, liposome-enabled nanotechnology still needs to be further optimized to realize its full therapeutic delivery potential.

While liposomes have shown potential to mitigate systemic toxicity associated with delivered therapeutic agents, systematic evaluation of side effects stemmed by liposomal nanocarriers in preclinical and clinical settings remains crucial [228,229]. Organ toxicity is a major concern of liposomal nanodrugs [230] because they prefer to accumulate in certain organs, such as the liver and spleen, affecting the tissue-specific functionality and potentially causing toxicities [231]. In addition, liposomes may interact with cell membranes, which can alter cell permeability and integrity, ultimately causing cellular damage [232]. Addressing these safety concerns requires strategic refinement and optimization. The formulation of liposomes plays a pivotal role in liposome-induced cytotoxicity. By further optimizing and modifying liposome composition, size, and surface properties, interactions with specific organs could be minimized. Additionally, efforts in enhancing the targeting specificity of liposomal dosage form are research directions that can greatly limit the off-target organ distribution, thus further reducing the systemic toxicities.

## 5. Conclusions

From the concept of liposomes being implemented for therapeutic applications to their recognition as the mainstream and most successful drug delivery platforms, the development path has been endless and tortuous in the past several decades. Nowadays, liposomes have been extensively used in cosmetics, dietetics, and pharmaceuticals, and most importantly in clinical applications for treating and managing a variety of diseases and conditions (e.g., cancers, infectious diseases, and pain) and for improved delivery of vaccines and gene therapeutics. Noteworthily, some studies have credited liposomal drugs more on reducing the side effects and toxicities than on increased efficacy compared with free drugs [233]. While free drug toxicities can be reduced by encapsulating into liposomes, the therapeutic effects are not bound to be improved in patients. Further optimization is required for liposomes; notwithstanding, significant clinical needs and challenges still await to be resolved in the future by liposomes.

## Figures and Tables

**Figure 1 pharmaceutics-16-00034-f001:**
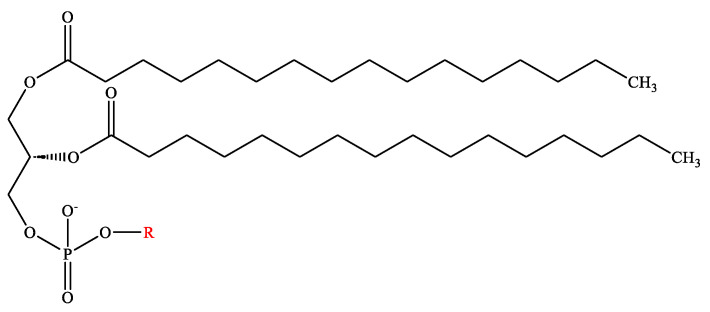
Chemical structure of phosphatidic acid (R = H). The red-labeled portion indicates the headgroup of the phosphatidic acid.

**Figure 2 pharmaceutics-16-00034-f002:**
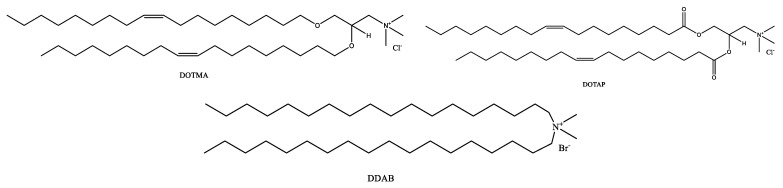
Chemical structures of DOTMA, DOTAP, and DDAB.

**Figure 3 pharmaceutics-16-00034-f003:**
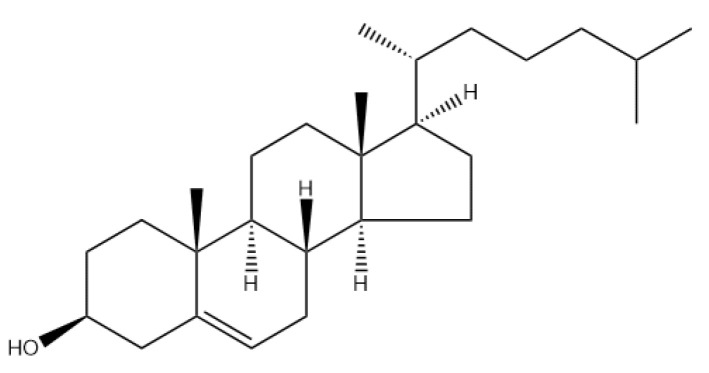
Chemical structure of cholesterol.

**Figure 4 pharmaceutics-16-00034-f004:**
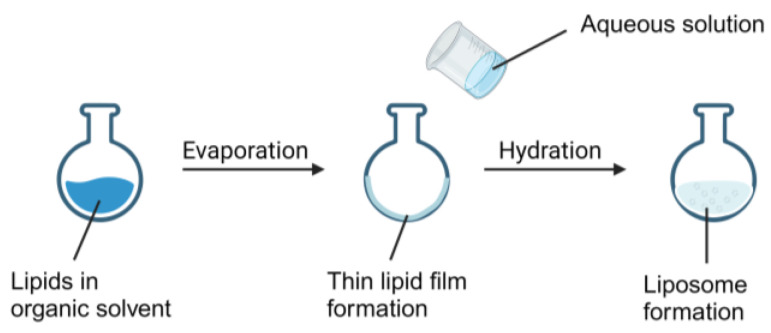
Demonstration of the use of the thin film hydration method to generate liposomes.

**Figure 5 pharmaceutics-16-00034-f005:**
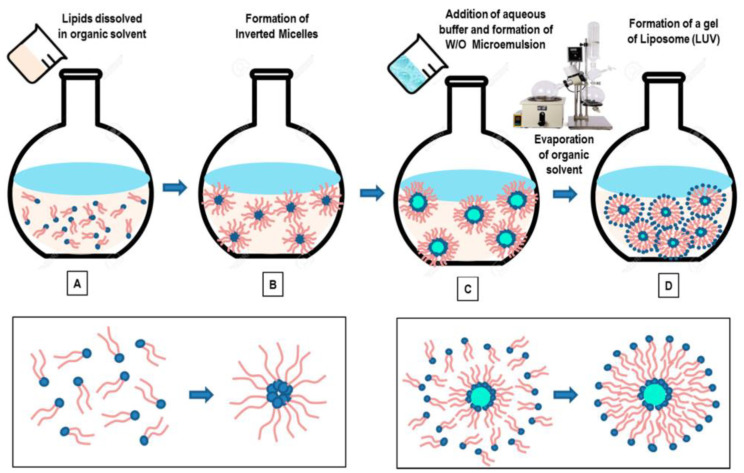
Illustration of the reverse phase evaporation method to produce liposomes [46]. Schematic representation of the main stages of the reverse-phase evaporation method. Lipids are dissolved in organic solvent (**A**), and the formation of inverted micelles is observed (**B**). The addition of aqueous media (buffer), followed by emulsification of the solution, favors the formation a homogeneous dispersion of a W/O microemulsion (**C**). With the final elimination of the organic solvent (by using rotary evaporation, under vacuum), a viscous gel is formed in the solution, which finally collapses to form liposomes (**D**) (LUVs).

**Figure 6 pharmaceutics-16-00034-f006:**
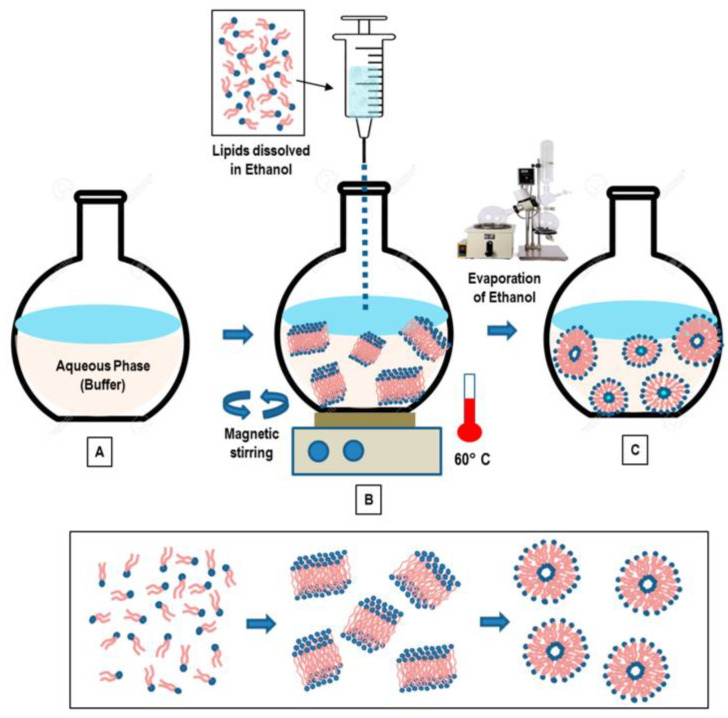
Solvent injection method for liposome preparation [46]. A composition of lipids dissolved in alcohol solution is injected into an aqueous phase (buffer) (**A**). The dilution of ethanol in the water solution favors the self-assembly of lipid components and the formation of bilayer planar fragments (**B**). Finally, the ethanol evaporation (depletion) favors the fusion of the lipids’ fragments and the formation of closed unilamellar vesicles (SUL and LUV) (**C**).

**Figure 7 pharmaceutics-16-00034-f007:**
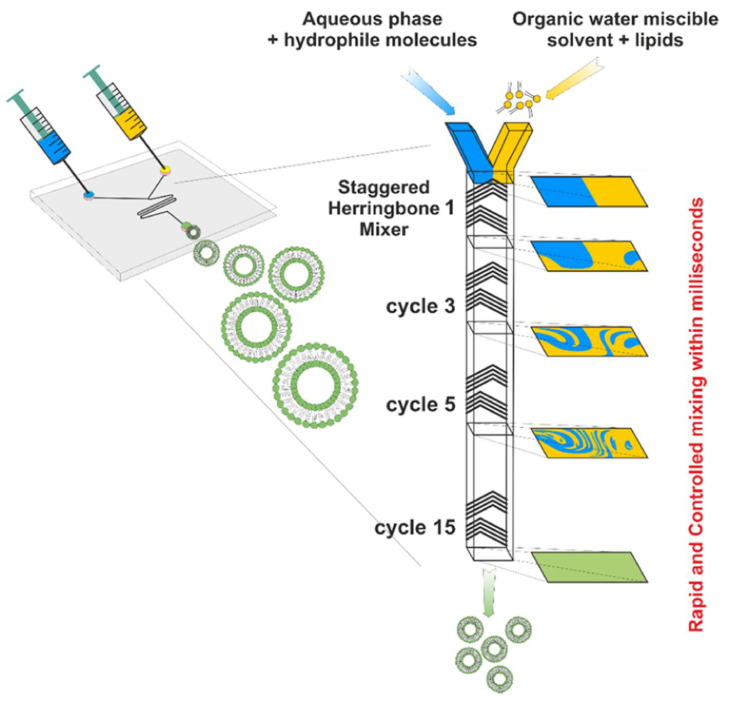
Illustration of microfluidic mixing and formation of liposomes [54].

**Figure 8 pharmaceutics-16-00034-f008:**
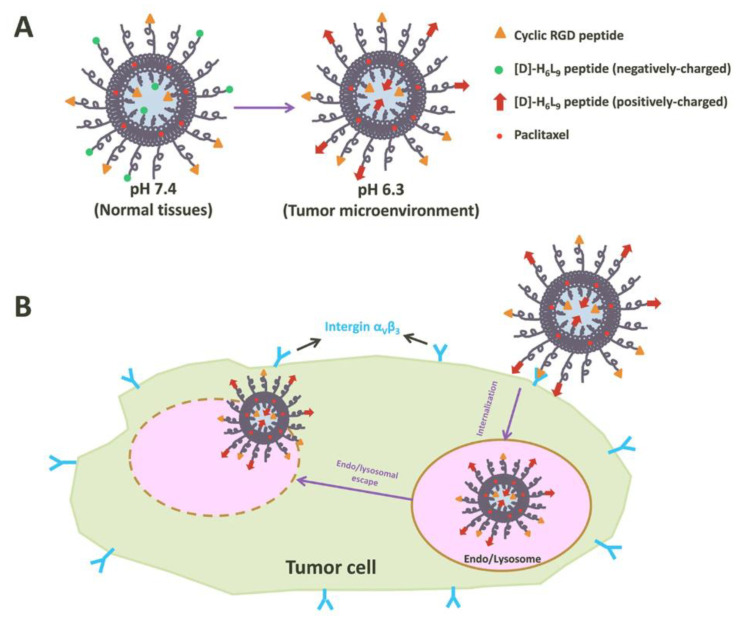
Illustration of tumor cell penetration with a peptide-decorated liposome [136]. (**A**) The Structure of peptide-decorated liposomes under different pH environments. (**B**) Within tumors, the peptide-decorated liposomes could target integrin α_V_β_3_ and initiate internalization and further intertumoral activities.

**Table 1 pharmaceutics-16-00034-t001:** Advantages of liposomes.

Advantages of Liposome:
▪ Improved solubility of encapsulated drugs
▪ Reduction of the free drug side-effects and toxicities
▪ Flexibility in size, charge, and lamellarity
▪ Non-toxic, biocompatible, and biodegradable
▪ Versatility with surface modifications

**Table 2 pharmaceutics-16-00034-t002:** Common phospholipids with their respective chemical formula of the headgroup.

Name of the Phospholipids	R-Group
Phosphatidic acid	-H
Phosphatidylethanolamine	-CH_2_-CH_2_-NH_3_^+^
Phosphatidylglycerol	-CH_2_-CHOH-CH_2_-OH
Phosphatidylcholine	-CH_2_-CH_2_-N(CH_3_)_3_^+^
Phosphatidylserine	-CH_2_-CH-NH_2_COOH
Phosphatidylinositol	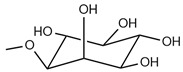

**Table 3 pharmaceutics-16-00034-t003:** U.S. FDA-approved liposomal/lipid-based nanodrugs.

Name	Clinical Approval Year	Liposomal Composition	Drug Encapsulated	Drug Type	Route of Administration	Company	References
Doxil	1995	HSPC:Cholesterol:DSPE-PEG2000	Doxorubicin	Chemotherapeutic	I.V.	Johnson & Johnson, Milpitas, CA, USA	[101,102]
Abelcet	1995	DMPC:DMPG	Amphotericin B	Antifungal	I.V.	Leadiant Biosciences. Inc., Rockville, MD, USA	[103,104]
DaunoXome	1996	DSPC:Cholesterol	Daunorubicin	Chemotherapeutic	I.V.	Galen US, Inc., Souderton, PA, USA	[101,105]
Amphotec	1996	Cholesteryl sulphate:Amphotericin B	Amphotericin B	Antifungal	I.V.	Sequus Pharmaceuticals Inc., Menlo Park, CA, USA	[101]
Inflexal V	1997	70% Lecithin, 20% Cephalin and 10% Phospholipids	Influenza virus antigen, strain A and B	Vaccine	I.M.	Sun Pharmaceutical Industries Ltd., Princeton, NJ, USA	[101,106]
Ambisome	1997	HSPC:DSPG:Cholesterol:Amphotericin B	Amphotericin B	Antifungal	I.V.	Fujisawa Healthcare, Inc. and Gilead Sciences, Inc., Foster City, CA, USA	[101]
Myocet	2000	EPG:Cholesterol	Doxorubicin	Chemotherapeutic	I.V.	Zeneus Pharma Ltd., Oxford, UK	[101,107]
Visudyne	2000	Verteporfin:DMPC and EPG	Verteporfin	Photosensitizer	I.V.	Novartis International AG, Basel, Switzerland	[101]
DepoDur	2004	DOPC:DPPG:Cholesterol:Tricaprylin and Triolein	Morphine sulfate	Narcotic Analgesic	Epidural	Pacira Pharmaceuticals, Inc., Watford, UK	[101,108]
Mepact	2004	DOPS:POPC	Mifamurtide	Immunomodulator/Antitumor	I.V.	Takeda Pharmaceutical Limited, Tokyo, Japan	[101]
Exparel	2011	DEPC:DPPG:Cholesterol:Tricaprylin	Bupivacaine	Anesthetic	I.V.	Pacira Pharmaceuticals, Inc., Parsippany-Troy Hills, NJ, USA	[101]
Onivyde	2015	DSPC:MPEG-2000:DSPE	Irinotecan	Chemotherapeutic	I.V.	Merrimack Pharmaceuticals, Inc., Cambridge, MA, USA	[101,109]
Vyxeos	2017	DSPC:DSPG:Cholesterol	Daunorubicin + Cytarabine	Antineoplastic	I.V.	Jazz Pharmaceuticals, Inc., Dublin, Ireland	[110]
Onpattro	2018	Cholesterol, DLin-MC3-DMA:DSPC:PEG2000-C-DMG	Patisiran	RNAi agent	I.V.	Alnylam Pharmaceuticals, Cambridge, MA, USA	[111]
Comirnaty	2021	ALC-0315:ALC-0159:cholesterol:DSPC	Nucleoside-modified mRNA encoding the viral spike (S) glycoprotein of SARS-CoV-2	Vaccine	I.M.	Pfizer-BioNTech, Mainz, Germany	[112]
Spikevax	2022	SM-102:mPEG2000-DMG:Cholesterol:DSPC	Nucleoside-modified mRNA encoding the viral spike (S) glycoprotein of SARS-CoV-2	Vaccine	I.M.	Moderna, Cambridge, MA, USA	[113]

## Data Availability

Data sharing is not applicable to this article as no new data was created or analyzed in this study.

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
