# Peer review of "Lipid-Based Nanotechnology: Liposome"

_pharmaceutics, 2023, doi:10.3390/pharmaceutics16010034_

Round 1
Reviewer 1 Report
Comments and Suggestions for Authors
The present state of biopharmaceutical chemistry—liposomal drug delivery systems—is the focus of the manuscript that is being presented. The review includes an overview of the procedures used to produce and characterizing liposomes, as well as the most recent information on FDA-approved medications based on liposomes.
However, a significant portion of the review's content consists of well-known information. I would like to add additional results for 2022 and 2023 to the review, greatly enriching it. A chapter containing liposomes covered with antimicrobial medications—including those for inhalation—must be added.
I suggest including information on fluorescence analysis and Fourier transform infrared spectroscopy in the section on liposome characterisation.
Author Response
Reviewer #1:
Q1. The present state of biopharmaceutical chemistry—liposomal drug delivery systems—is the focus of the manuscript that is being presented. The review includes an overview of the procedures used to produce and characterizing liposomes, as well as the most recent information on FDA-approved medications based on liposomes. However, a significant portion of the review's content consists of well-known information. I would like to add additional results for 2022 and 2023 to the review, greatly enriching it. A chapter containing liposomes covered with antimicrobial medications—including those for inhalation—must be added.
Response: Thank you for the comment. This section has been added to “3.2.3 Inhaled liposomal antimicrobial medications”, shown as followed (line 567-584):
3.2.3. Inhaled liposomal antimicrobial medications
Besides the most common routes of liposomal drug delivery, such as oral, typical, and parenteral, the use of inhaled liposomal dosage form for treating respiratory diseases and/or infections has been used increasingly in clinical practice (Krishna et al., 2023). Inhaled liposomal dosage forms offer potential advantages for treatment of respiratory infections including targeted delivery, reduced systemic toxicity, improved efficacy, and minimized side effects (Dhiman et al., 2022; Forest & Pourchez, 2022; Vuong et al., 2023).
Amphotericin B, as the common treatment for pulmonary fungal infections, is limited with high mortality in achieving the minimum inhibitory concentration in the lung (Alabdullah & Yousfan, 2023). To address these challenges, clinical studies have explored the administration of liposomal amphotericin B parenteral formulation alone or with amphotericin B deoxycholate through nebulization (Muthu et al., 2023). Nebulization is the method that converts medications into fine mist for inhalation, which has shown promise in enhancing the delivery of amphotericin B and combating pulmonary fungal infections. Apart from the nebulization, pressurized metered-dose inhalers (pMDIs) and dry powder inhalers (DPIs) are also being used for drug delivery, but each device requires different formulations to ensure successful drug delivery into the lung (Celi et al., 2023; de Pablo et al., 2023; Saalbach, 2023).
Q2. I suggest including information on fluorescence analysis and Fourier transform infrared spectroscopy in the section on liposome characterisation.
Response: Thank you for the suggestion. Requested sections have been added to the manuscript, “2.5.4. Fluorescence microscopy”, and “2.5.5. Fourier transform infrared spectroscopy”, as following (line 344-370):
2.5.4. Fluorescence microscopy
Fluorescence microscopy techniques enable multi-color imaging of individual liposome, providing a more quantitative and comprehensive understanding of their biochemical properties (Chen et al., 2023). Among these techniques, two commonly utilized methods include laser scanning confocal fluorescence microscopy (LSCFM) and total internal reflection fluorescence microscopy (TIRFM). LSCFM, known for its high resolution, optical sectioning, and 3D imaging capabilities (Dey et al., 2022; Oleksiievets et al., 2022), serves to characterize liposomes, observes their structure, behavior, and interactions with high precision (Ciobanasu, 2022; Karakas et al., 2023). In contrast, TIRFM is well-known for its accessibility and high sensitivity in probing biomolecules properties (Colson et al., 2023). When characterizing liposome, TIRFM presents unique advantages by allowing observations of membrane-related phenomena and interactions at the liposome interface (Ashby et al., 2023; Scheeder et al., 2023). Both LSCFM and TIRFM play pivotal roles in unraveling the liposomal structure and their interactions, contributing significantly to our understanding of the liposomal delivery platforms.
2.5.5. Fourier transform infrared spectroscopy
Fourier transform infrared spectroscopy (FTIR) is a potent analytical technique extensively utilized in liposomes research, including composition assessment, structural analysis, interaction studies, and stability evaluation (Goh & Lane, 2022; Meng et al., 2023; Naeeni et al., 2023). Different functional groups in lipids have unique peaks in the FTIR spectrum, enabling the straightforward determination of lipid composition and ratios within liposomes. Changing in specific absorption bonds will be indicated as interactions between lipid and encapsulated molecules, such as hydrogen bonding, while also serving as a mean to access liposome stability by detecting alteration because of oxidation or degradation. Thus, FTIR provides invaluable insight into the liposome characterization, contributing significantly to the optimization of liposome-based drug delivery platforms.

Reviewer 2 Report
Comments and Suggestions for Authors
Including illustrations of the major methods of preparation of liposomes will make the review more attractive to the readers.
The paper covers several areas of the production and characterization of liposomes but a more in-depth discussion must be provided. The review as presented is too superficial in several parts.
Stimuli-responsive liposomes should be discussed in the paper, such as pH, temperature, etc.
Toxicological issues and analysis of systemic toxicity in preclinical and clinical trials should be discussed.
Author Response
Reviewer #2:
Q1. Including illustrations of the major methods of preparation of liposomes will make the review more attractive to the readers.
Response: Thank you for the comment. Additional illustrations for liposome preparation techniques have been added as followings:
Figure 4. Demonstration of the use of the thin film hydration method to generate liposomes (Gharib et al., 2015).
Figure 5. Illustration of reverse phase evaporation method to produce liposomes (Gharib et al., 2015).
Figure 6. Solvent injection method for liposome preparation (Gharib et al., 2015).
Q2. The paper covers several areas of the production and characterization of liposomes but a more in-depth discussion must be provided. The review as presented is too superficial in several parts. Stimuli-responsive liposomes should be discussed in the paper, such as pH, temperature, etc.
Response: Thank you for the comment. To have a more in-depth discussion of liposomes, we have added “2.5.4. Fluorescence microscopy”, and “2.5.5. Fourier transform infrared spectroscopy” (line 344-370). Also, as suggested, “2.6. Stimuli-responsive liposomes” section has been added as following: (line 372-426):
2.6. Stimuli-responsive liposomes
Stimuli-responsive liposomes are specifically designed to release their payloads in response to specific external stimuli. Many environmental stimuli have been explored including pH, temperature, and enzymes.
2.6.1. pH-responsive liposomes
pH-responsive liposomes are specialized nanocarriers that are responsive to alternations in the acidity or alkalinity of the surroundings. These liposomes are engineered to maintain their stability at certain pH and then can rapidly dissociate once exposed to a different pH environment (AlSawaftah et al., 2022). Normally, most of the components in pH-responsive liposomes are similar to other liposomal drugs, except for the pH-sensitive component. Zhao et al. developed a pH-responsive liposome platform for co-delivery of PLK-1 specific siRNA and docetaxel using a dual pH-sensitive peptide sequence, composed of three unites: a cell-penetrating domain (polyarginine), a polyanionic shielding domain (ehG)n, and an imine linkage (Zhao et al., 2022). During the blood circulation, the liposomal formulation remains inert until reaching the acidic tumor microenvironment, and then its contents release is initiated by the breakage of the imine bonds within the pH-responsive peptide under acid-catalyzed hydrolysis.
Additionally, pH-responsive liposomes have been investigated for the delivery of various therapeutic agents, such as proteins, nucleic acids, and chemotherapeutics drugs (Bami et al., 2022; Shinn et al., 2022).
2.6.2. Temperature-responsive liposomes
Temperature-responsive liposomes can alter their original structures or behavior in response to varied temperature. These liposomes are designed to maintain their stability at one temperature and then undergo conformational changes at a different temperature threshold.
Zhao et al. investigated a temperature-responsive liposome based on the insertion of the ion pair of polyethyleneimine (PEI) and (phenythio)acetic acid (PTA) into the liposomal bilayer (Zhao et al., 2023). This ion pair, formed by electrostatic interaction between PEIs and PTAs, possesses amphiphilic properties, facilitating self-assembling in the aqueous solution (Kim et al., 2021). Being a compatible molecule, the amphiphilic ion pair contributes to the creation of DOPE liposomes, with the PTAs' phenyl head group inserting into the bilayer while PEI's chain fills the space among the phospholipid headgroups. As the temperature rises, the ion pair loses its amphiphilicity due to the hydration of the phenyl group on PTAs (Zhao & Kim, 2022). This will enable the detachment of the ion pair from DOPE liposomes, destabilizing the liposomes and consequently releasing the payload.
Temperature-sensitive liposomes provide the advantage of precise control over drug release triggered by temperature change, which has been widely applied in various cancer treatments (Razmimanesh & Sodeifian, 2023; Q. Yang et al., 2023).
2.6.3. Enzyme-responsive liposomes
Enzyme-sensitive liposomes are designed to release their encapsulated payloads in the presence of specific enzymes. The controlled release of their contents is triggered by the exposure or recognition of particular enzymes.
Chasteen et al. explored an enzyme-responsive liposomes by modifying the N-acylated DOPE into the liposomes (Chasteen et al., 2023). Exposure to specific temperature (55 °C to 65 °C) in the presence of palladium, a catalyst for biorthogonal chemistry (Sun et al., 2022), it will generate triggered-release liposomes involves crafting chemically caged polar lipids capable of transitioning from a structure that maintain a liposomal membrane to one that disrupts the bilayer.
Enzyme-responsive liposomes show potentials for targeted drug delivery in various medical fields, as they could be easily tailored to respond to different enzymes associated with specific diseases for precise therapeutic delivery (S. Wang et al., 2023; Q. Yang et al., 2023).
Q3. Toxicological issues and analysis of systemic toxicity in preclinical and clinical trials should be discussed.
Response: Thank you for the comment. The "toxicological issues and analysis of systemic toxicity” has been added to the “4. Discussion and Outlook” section (line 731-744) as following:
While liposomes have shown potential to mitigate systemic toxicity associated with delivered therapeutic agents, systematic evaluation of side effects stemmed by liposomal nanocarriers in preclinical and clinical settings remains crucial (Aloss & Hamar, 2023; Kommineni et al., 2023). Organ toxicity is a major concern of liposomal nanodrugs (de Oliveira Silva et al., 2023) because they prefer to accumulate in certain organs, such as liver and spleen, affecting the tissue-specific functionality and potentially causing toxicities (Oros-Pantoja et al., 2022). In addition, liposomes may interact with cell membranes, which can alter cell permeability and integrity, ultimately causing cellular damage (Ranjbar et al., 2023). Addressing these safety concerns requires strategic refinement and optimization. The formulation of liposomes plays a pivotal role in liposome-induced cytotoxicity. By further optimizing and modifying liposome composition, size, and surface properties, interactions with specific organs could be minimized. Additionally, efforts in enhancing the targeting specificity of liposomal dosage form are research directions that can greatly limit the off-target organ distribution, thus further reducing the systemic toxicities.

Reviewer 3 Report
Comments and Suggestions for Authors
The review paper is not well written. It raises too many issues such as: classification of liposomes, their physicochemical properties, methods of their production and selected applications. Such many issues discussed means that they are only briefly presented. Therefore, the work is only a very superficial description of a large area of knowledge. It makes the reader dissatisfied due to the lack of comprehensive descriptions and analyses. There is no leading theme in this review.
Additionally, in many places the terms are used incorrectly or are not well defined. Below are some examples:
Line 83 It is not clear why fluidity has been selected as a relevant parameter whereas surface electrical potential and hydrophobicity or propensity for hydrogen bond formation are omitted?
Line 87 “….one drop….” as a quantitative value is not very precise and to collcvial.
Line 94 It Is not clear how the phosphatidic acid is understood; as lipid molecule or as a component of a phospholipids. As lipid molecule it is not the most abundant phospholipid.
Line 106 …. solubilization and metabolism of what?
The fragment from line 123 to line 131 is misleading, cholesterol is not a “cement” required for lipid bilayer formation. Phospholipids by themself without cholesterol will form liposomes in water. The hydrophobic effect is a driving force of liposome formation, the paragraph need to be rewritten. Moreover, cholesterol will not form liposome by itself.
The chapter 2.3 PEG should be reorganized. At current state it is confusing. Authors write about PEG modification of biologically active compounds and later about surface modification of liposomes. There is no logical connection between the two. EPR is not an effect of the presence of PEG on liposome surface. It is a result of liposome size.
Line 164 - the freezing-drying procedure is usually performed on hydrated lipids.
Line 167 why rehydration, hydration will suffice in the context?
Line 168 the efficient encapsulation in this case does not depend on liposome formation temperature.
Lines 172 – 175 Statements in this section are not clear.
Line 192 “…results in larger aqueous phase ….“ refers to entrapped aqueous phase ???
Figure 4 does not show how liposomes are formed in microfluidic device.
Line 264 Liposome cannot have 10 nm size. A single lipid blayer is 5 nm thick.
Line 270 liposomes cannot accumulate in proteins.
Line 282 The leakage of active compound does not depend exclusively on liposome properties the chemical nature of the drug is also relevant.
Chapter 253
Lipid aggregates have two types of phase transitions lyotropic and thermotropic.
Figure 5
There are number of erroneous concepts in Figure 5.
The image if inverted micelles is not correct. The liposome fusion with the plasma membrane in most cases is not a spontaneous process nor the frequent one.
In summary, the text requires major revisions and should not be published in the current form.
Comments on the Quality of English LanguageThe text requires major language revision. Many terms are not correctly used.
Author Response
Reviewer #3:
Q1. The review paper is not well written. It raises too many issues such as: classification of liposomes, their physicochemical properties, methods of their production and selected applications. Such many issues discussed means that they are only briefly presented. Therefore, the work is only a very superficial description of a large area of knowledge. It makes the reader dissatisfied due to the lack of comprehensive descriptions and analyses. There is no leading theme in this review. Additionally, in many places the terms are used incorrectly or are not well defined. Below are some examples:
Line 83 It is not clear why fluidity has been selected as a relevant parameter whereas surface electrical potential and hydrophobicity or propensity for hydrogen bond formation are omitted?
Response: : Thank you for the comment. As suggested, the “fluidity” parameter has been deleted.
The relevant statement has been revised as following (line 83-84):
Three dominant components that contribute to formation, stability, and functionality of liposomes include phospholipids, cholesterol, and polyethylene glycol (PEG).
As suggested, “Electrical potential” is discussed and added in “2.5.1. Size distribution and zeta-potential” as following (line 277-282):
Liposomes with positive surface charge facilitate the absorption into tissues, whereas those with a slightly negative charge appear to prolong the circulation time by efficiently minimizing the protein binding and opsonization. Thus, we can adjust the surface charges on liposomes to potentially improve in vivo targeting efficacy (Beltrán-Gracia et al., 2019; Ren et al., 2019). Additionally, liposomes with neutral surface charge tend to aggregate (Large et al., 2021).
As suggested, “Propensity for hydrogen bond formation" is discussed and added in “2.5.2. Stability and drug leakage” as following (line 310-319):
The propensity of hydrogen bond formation in liposomes is crucial for structural stability. Hydrogen bonds serve as a pivotal factor in maintaining the structure integrity of liposomes (Haneef et al., 2023). These bonds are conducive to stabilizing the phospholipid bilayer, which constitutes the basic structure of liposomes, by enhancing interaction between lipid molecules. This could prevent the breakdown and sustain the integrity of the vesicle structure. Moreover, it also plays a vital role in membrane permeability (Raju et al., 2023). Depending on the types or strength of hydrogen bonds present, the liposome membrane displays varied degrees of permeability. Consequently, it is possible to regulate the encapsulation and subsequent drug release within the liposomes via modulating the hydrogen bonding.
Q2. Line 87 “….one drop….” as a quantitative value is not very precise and to collcvial.
Response: Thank you for the comment. As suggested, this fraction of the sentence has been removed from the text and now it reads as following (line 86-89):
In the aqueous solution, phospholipids self-assemble into a bilayer sheet where the headgroups align, forming two lipid layers with their tail groups facing each other (Liu et al., 2020).
Q3. Line 94 It Is not clear how the phosphatidic acid is understood; as lipid molecule or as a component of a phospholipids. As lipid molecule it is not the most abundant phospholipid.
Response: Thank you for the comment. The statement in the text has been rewritten as following (line 94-96):
Phosphatidic acid (PA) is a phospholipid found in cell membranes and acts as a building block for the biosynthesis of other phospholipids (Fig. 1).
Q4. Line 106 …. solubilization and metabolism of what?
Response: Thank you for the comment. The statement has been revised as followed (line 106-110):
It also acts as a surfactant and helps in solubilization and metabolism of the vesicles, as phospholipid could not only maintain the integrity of cellular structure and functions to regulate the passage of molecules, ions, and nutrients in/out of the cells, but contribute to forming lipid droplets as triglycerides to store energy (Jin et al., 2023; Van Hoogevest & Wendel, 2014; Waghule et al., 2022).
Q5. The fragment from line 123 to line 131 is misleading, cholesterol is not a “cement” required for lipid bilayer formation. Phospholipids by themself without cholesterol will form liposomes in water. The hydrophobic effect is a driving force of liposome formation, the paragraph need to be rewritten. Moreover, cholesterol will not form liposome by itself.
Response: Thank you for the comment. The statement has been rewritten as following (line 126-135):
Additionally, within the phospholipid bilayer, cholesterol significantly influences membrane fluidity, permeability, and stability by interacting with the nonpolar tails of phospholipids and their polar headgroups. The interaction of cholesterol involves binding its nonpolar portion to the hydrophobic tails of phospholipids while its polar head group associates with the hydrophilic headgroups of phospholipids (Albuquerque et al., 2018; Cerqueira et al., 2016). Consequently, cholesterol plays a dual role in contributing to the formation of phospholipid bilayers and impacting the properties of liposomes. Also, it impacts the rigidity, strength, and permeability of the lipid membrane by modulating the ratio of phospholipids to cholesterol within the bilayers.
Q6. The chapter 2.3 PEG should be reorganized. At current state it is confusing. Authors write about PEG modification of biologically active compounds and later about surface modification of liposomes. There is no logical connection between the two. EPR is not an effect of the presence of PEG on liposome surface. It is a result of liposome size.
Answer: Thank you for the comment. As suggested, PEG modification of biologically active compounds has been deleted and the Section 2.3 PEG has been rewritten as following (line 138-151):
2.3. Polyethylene glycol
Polyethylene glycol (PEG) forms through the polymerization of ethylene glycol, ethylene oxide, or diethylene glycol in the presence of water and a catalyst under pressure (J. Brady et al., 2017; Shi et al., 2022). The chemical structure of PEG generally comprises OH-[CH2-CH2-O]n-H. Being hydrophilic in nature, PEG can be covalently attached to the surface of liposome to impart the “stealth” effect (Wang et al., 2022; Woodle et al., 1992) via educing protein binding and opsonization as well as diminishing degradation by metabolic enzymes. This occurs because PEG prevents the approach or recognition by blood proteins, proteolytic enzymes and antibodies during circulation, thereby enhancing the therapeutic index in comparison to the raw drugs (Fischer et al., 2019; Wang et al., 2020). Notably, PEGylated liposomes have been shown to increase in vivo circulation time and improved formulation stability (Mohamed et al., 2019). This extension of circulation time and enhancement of stability are crucial factors in improving the performance of liposomal drug delivery systems.
Q7. Line 164 - the freezing-drying procedure is usually performed on hydrated lipids.
Response: Thank you for the comment. As advised, the relevant statement has been revised as following (line 164-165):
A freezing-drying procedure, typically conducted on hydrated lipids, can also be utilized to ensure the complete removal of solvents.
Q8. Line 167 why rehydration, hydration will suffice in the context?
Response: Thank you for the comment. To make it clearer, the relevant sentence has been revised as following (line 166-167):
This step facilitates the encapsulation of drugs into the lipid thin film, and subsequent hydration triggers the formation of liposome.
Q9. Line 168 the efficient encapsulation in this case does not depend on liposome formation temperature.
Response: Thank you for the comment. This statement has been rewritten as following (line:167-169):
It is also crucial that the temperature of the hydration buffer remains above the gel-lipid phase transition temperature (Tm) (Large et al., 2021).
Q10. Lines 172 – 175 Statements in this section are not clear.
Response: Thank you for the comment. To make it clearer, this section has been rewritten as following (line 169-173):
Furthermore, several other critical factors affecting liposomal characteristics include the volume of the hydration solution and the rate of hydration (Reeves & Dowben, 1969; Vemuri & Rhodes, 1995). A large volume of the hydration solution tends to generate a great number of multilamellar vesicles (MLVs) with a more heterogeneous size distribution.
Q11. Line 196 “…results in larger aqueous phase ….“ refers to entrapped aqueous phase ???
Response: Thank you for the comment. To make it clearer, this section has been rewritten as following (line 194-197):
Reverse phase evaporation can generate a mixture of LUVs and MLVs, resulting in the entrapped aqueous phase. This outcome significantly enhances drug loading capacity and allows for the entrapment of larger molecules, such as proteins or nucleic acids (Akbarzadeh et al., 2013).
Q12. Figure 4 does not show how liposomes are formed in microfluidic device.
Response: Thank you for the comment. The Figure 4 has been replaced with a new microfluidic illustration (now Figure 7) that shows how liposomes are formed.
Figure 7. Illustration of microfluidic mixing and formation of liposomes (Kotouček et al., 2020).
Q13. Line 264 Liposome cannot have 10 nm size. A single lipid blayer is 5 nm thick.
Response: Thank you for spotting this error. This section has been rewritten as following (line 271-274):
Liposome sizes range from very small (0.025µm) to large (2.5µm), typically designed to be <400 nm to enable passive accumulation of liposomes within the tumor microenvironment through the enhanced permeation and retention (EPR) effect (Akbarzadeh et al., 2013; Subhan et al., 2021).
Q14. Line 270 liposomes cannot accumulate in proteins.
Response: Thank you for the comment. As suggested, we have revised the relevant statement as following (line 276-281):
Liposomes with positive surface charge facilitate the absorption into tissues, whereas those with a slightly negative charge appear to prolong the circulation time by efficiently minimizing the protein binding and opsonization. Thus, we can adjust the surface charges on liposomes to potentially improve in vivo targeting efficacy (Beltrán-Gracia et al., 2019; Ren et al., 2019).
Q15. Line 282 The leakage of active compound does not depend exclusively on liposome properties the chemical nature of the drug is also relevant.
Response: Thank you for the comment. As suggested, relevant information has been added to “2.5.2 stability and drug leakage” section, as following (line 301-308):
Additionally, instability of liposomes induced by the chemical nature of the encapsulated drugs also remain a concern in drug delivery. The interactions between drugs and phospholipids can disrupt chemical stability of liposomes due to various drug-related factors, including hydrophobicity/hydrophilicity, pH sensitivity, and chemical reactivity (Guimarães et al., 2021; Jyothi et al., 2022; Khan et al., 2008). Hence, understanding the inherent chemical properties of the delivered drug is crucial in designing liposomal formulations that minimize leakage and improve drug retention within the vehicles.
Q16. Chapter 253 Lipid aggregates have two types of phase transitions lyotropic and thermotropic.
Response: Thank you for the comment. As suggested, the information about the lyotropic transition has been added to “2.5.3 Phase Transition” section, shown as followed (line 327-341):
Additionally, lipid aggregates undergo two distinct types of phase transition: thermotropic and lyotropic (Smaisim et al., 2023).
Thermotropic phase transitions occur when temperature changes, leading to alterations in the organization and composition of lipid structures. Lipid molecules can transition between various states, such as from a structured and rigid gel phase to a more fluid and less ordered liquid crystalline phase as temperature increases. These changes relate to the packing and mobility of lipid molecules within the assemblies as temperature fluctuates. (Kianfar et al., 2020; Smaisim et al., 2023).
Changes in solvent concentration, such as water, within a system containing lipids or amphiphilic compounds lead to lyotropic phase transitions. These transitions involve restructuring and altering the structure of lipid aggregates, such as micelles or bilayers, in response to changes in solvent concentration. For instance, fluctuations in water concentration prompt lipid molecules to adopt various organized arrangements like lamellar structures (sheets), hexagonal patterns, or micelles. These modifications occur due to interactions between the lipids and the solvent molecules. (Smaisim et al., 2022).
Q17. Figure 5 There are number of erroneous concepts in Figure 5.
The image if inverted micelles is not correct. The liposome fusion with the plasma membrane in most cases is not a spontaneous process nor the frequent one.
Response: Thank you for the valuable comment. To avoid the confusion, the Figure 5 has been replaced with the following illustration:
Figure 5. Illustration of tumor cell penetration with a peptide-decorated liposome (Zhang et al., 2016).
Q18. In summary, the text requires major revisions and should not be published in the current form.
Response: Thank you for the comment. As suggested, the text has been revised comprehensively.

Round 2
Reviewer 1 Report
Comments and Suggestions for Authors
The authors have improved the manuscript and now it is suitable for the acceptance
Reviewer 2 Report
Comments and Suggestions for Authors
The manuscript was improved by the authors and it can be accepted
Reviewer 3 Report
Comments and Suggestions for Authors
After corrections and addition the review is suitable for publication.